# Real-Life Pre-Operative Nodal Staging Accuracy in Non-Small Cell Lung Cancer and Its Relationship with Survival

**DOI:** 10.3390/diagnostics15040430

**Published:** 2025-02-11

**Authors:** Ahmed Alkarn, Liam J. Stapleton, Dimitra Eleftheriou, Laura Stewart, George W. Chalmers, Ahmad Hamed, Khaled Hussein, Kevin G. Blyth, Joris C. van der Horst, John D. Maclay

**Affiliations:** 1Glasgow Royal Infirmary, Glasgow G4 0SF, UK; ahmed.alkarn@mft.nhs.uk (A.A.); liamstapleton@nhs.net (L.J.S.); george.chalmers@nhs.scot (G.W.C.); joris.vanderhorst@nhs.scot (J.C.v.d.H.); 2Faculty of Medicine, Assiut University, Assiut 71515, Egypt; prof.ahmad.hamed.assiut@gmail.com (A.H.); khaldhussein@yahoo.com (K.H.); 3Department of Statistics, University of Glasgow, Glasgow G4 0SF, UK; d.eleftheriou@lacdr.leidenuniv.nl (D.E.); laura.stewart@ed.ac.uk (L.S.); 4Institute of Cancer Sciences, University of Glasgow, Glasgow G4 0SF, UK; kevin.blyth@glasgow.ac.uk; 5Glasgow Pleural Disease Unit, Queen Elizabeth University Hospital, Glasgow G4 0SF, UK

**Keywords:** non-small cell lung cancer, mediastinal staging, endobronchial ultrasound (EBUS), thoracic surgery, survival

## Abstract

**Background/Objectives:** The precise staging of non-small cell lung cancer (NSCLC) determines its initial treatment and provides more accurate prognostic information for patients. The aim of this cohort study was to determine pre- and post-operative mediastinal nodal staging agreement and its effect on 2-year survival. **Methods:** A retrospective multi-centre cohort study was performed, using prospectively collected and pre-defined data from weekly lung cancer multidisciplinary team (MDT) meetings in 11 hospitals. Consecutive patients who underwent surgical resection of NSCLC between 2015 and 2017 were eligible. Pre-operative under-staging was defined as a lower pre-operative than post-operative nodal stage, and pre-operative over-staging as a higher pre-operative than post-operative nodal stage. Disparities between pre-operative nodal staging established at MDT and post-surgical nodal staging were determined and associations with subsequent lung cancer-specific 2-year mortality were identified using univariate and multivariate regression. **Results:** A total of 973 patients fulfilled the eligibility criteria. Concordant pre- and post-operative nodal staging was observed in 783/973 (80%), 123/973 (13%) were under-staged pre-operatively and 67/973 (7%) were over-staged. In 173 patients with clinical N1 or N2 disease (in whom invasive mediastinal staging was indicated), staging EBUS was performed in 55/173 (32%). In these patients, younger age and use of staging EBUS were independent predictors of concordant pre- and post-operative staging. In all patients, pre-operative under-staging was independently associated with increased lung cancer-specific 2-year mortality. There was no increased mortality associated with pre-operative nodal over-staging. **Conclusions:** Invasive mediastinal staging with EBUS was independently associated with more accurate pre-operative staging. Pre-operative nodal under-staging was associated with increased lung cancer-specific mortality. Nodal staging accuracy in potentially curable NSCLC is of fundamental importance to ensure patients receive the correct first-line treatment and to improve survival.

## 1. Introduction

Although the national and international guidelines for the diagnosis and management of lung cancer differ, they all mandate mediastinal staging with endobronchial ultrasound (EBUS), endoscopic ultrasound (EUS) and/or mediastinoscopy for non-small cell lung cancer (NSCLC) if any intrathoracic node is 10 mm or greater or shows FDG uptake on PET-CT scanning [1,2,3].

In the UK, pre-surgical induction chemotherapy is not common practice due to concerns regarding patients’ subsequent fitness for surgery and a similar survival benefit from adjuvant chemotherapy [4]. As such, a common suggestion is that multimodal treatment could be determined by post-operative stage. However, the recent publication of the CheckMate 816 study has fundamentally changed the landscape of treatment for resectable but locally advanced NSCLC [5]. Pre-treatment staging is of fundamental importance to select patients suitable for neoadjuvant chemotherapy plus nivolumab followed by surgery, which confers a significant survival benefit in stage IB-3 disease, irrespective of PD-L1 status. Furthermore, in patients with unresectable stage 3 disease, the PACIFIC study demonstrated a survival benefit in treating stage 3 patients with concurrent chemoradiotherapy plus durvalumab [6]. As such, ensuring the pre-treatment stage is correct will ensure that these patients embark on the correct treatment path from the outset.

Comparing pre-operative clinical staging with post-operative surgical staging based on lymph node resection allows the accuracy of the pre-treatment nodal staging work-up to be objectively assessed. More importantly, such comparisons also allow the impact of discordant pre-operative staging to be quantified and the factors associated with this identified and addressed. A recent Dutch audit reported an agreement between pre- and post-surgical nodal staging of 79% [7]. In recent years, staging EBUS +/− EUS alone has been used in place of mediastinoscopy and has been shown to be non-inferior to both procedures combined in diagnosing unforeseen N2 disease with less morbidity [8].

The aim of this retrospective cohort study was to determine the level of concordance between pre- and post-operative mediastinal nodal staging in a cohort of surgically resected NSCLC patients assessed using modern staging investigations. We also sought to identify any impact of discordant nodal staging on lung cancer-specific 2-year mortality.

## 2. Patients and Methods

### 2.1. Patients

Consecutive patients diagnosed in the West of Scotland (a region including 11 centres with a mix of both university teaching and district general hospitals comprising 7 multidisciplinary teams) were included. Cases were eligible if they were diagnosed with NSCLC between 1 January 2015, and 31 December 2017 and underwent surgical resection as first treatment. Cases were excluded if pre-operative staging was not recorded and/or no PET-CT was performed (Figure 1). These data were prospectively collected locally by clinical audit staff in each NHS Board from diagnosis to definitive treatment in accordance with the nationally agreed-upon Quality Performance Indicator dataset and definitions, and storage of these data for future analysis is approved nationally. These routine data were then matched with cause-of-death data from death certification held by NHS National Services Scotland. The Caldicott Guardian oversees the storage and use of these patient identifiable data for audit and research purposes. Permissions for specific analyses were sought a priori and were approved by the local Caldicott Guardian. Patients’ electronic clinical records were reviewed to identify lung cancer recurrence in all patients at two years post-surgery. In patients deemed to have died of other causes, electronic clinical records were reviewed to confirm this. Patients who died either during their admission for lung cancer resection or within 30 days of their discharge were excluded from the survival analyses. Details regarding invasive mediastinal staging and any missing data were extracted from electronic patient records. Performance status was defined according to the Eastern Cooperative Oncology Group scale [9]. For statistical analyses, patients were classified into 3 histopathological groups: squamous cell carcinoma, adenocarcinoma and other. Wait-time until surgery was defined as the number of days between the first radiological diagnosis by CT and resection. The follow-up period for survival was 2 years from the date of surgery.

The pre-operative stage was defined as the stage determined prior to surgery after all pre-operative staging investigations were complete, including PET-CT and any invasive mediastinal staging procedures e.g., EBUS, EUS or mediastinoscopy, usually established at multi-disciplinary team meetings. In general, only patients with N0, N1 or single-station N2 disease were deemed suitable for surgery. Post-operative stage was based on the pathological examination of the resected tumour and lymph nodes. Patients were staged using the TNM Classification of Malignant Tumours, 7th Edition [10]. Concordant (or accurate) staging was defined as identical pre- and post-operative nodal stages. Regarding discordant or inaccurate staging, pre-operative under-staging was defined as a lower pre-operative than post-operative nodal stage and pre-operative over-staging was defined as a higher pre-operative than post-operative nodal stage.

### 2.2. Statistical Analyses

The data are reported as simple proportions (%), mean (SD) if normally distributed or median (IQR) if not. A univariate analysis for factors associated with discordant clinical N staging was performed in a pre-planned sub-group analysis of patients with pre-operative N1 and N2, in whom invasive mediastinal staging was indicated according to international guidelines [1,2,3], using Chi-squared and Mann–Whitney U tests as appropriate; variables with a *p* < 0.1 were subsequently entered into a binary logistic regression analysis focussed on the same outcome measure.

A competing relative risk analysis (Fine and Gray) was performed to examine cancer-specific 2-year mortality in all patients, as a proportion of patients in our cohort had died of non-cancer-related causes [11]. This allowed us to determine the factors independently associated with mortality, including nodal staging concordance and post-operative staging, presented with hazard ratios with 95% confidence intervals.

IBM^®^ SPSS^®^ Statistics 25.0 (Armonk, NY, USA) and R 4.1.1 (Vienna, Austria) were used for the statistical analyses. *p* < 0.05 was considered to indicate significance.

## 3. Results

The study flowchart is presented in Figure 1. A total of 973 patients with clinical stage IA to IIIB NSCLC underwent surgical resection as an initial treatment between 2015 and 2017 and fulfilled all eligibility criteria. The exclusions are documented in Figure 1. Eighteen patients were excluded from the survival analyses as they died in the immediate post-operative period. The baseline characteristics of the study population are summarized in Table 1. The mean (SD) age was 69 (8.7), and there was a slight female predominance (Table 1).

### 3.1. Accuracy of Pre-Operative Nodal (N) Stage Compared to Post-Operative (N) Stage

Pre- and post-operative N stages were concordant in 783/973 of patients (80%, Table 2). Pre-operative stage N0 was associated with the most accurate pre-operative staging, with 89% concordance, in comparison to stages N1 and N2 (37% and 58%, respectively). There was a higher rate of pre-operative nodal under-staging in patients undergoing pneumonectomy than in those treated by lobectomy or sublobar resection (40% vs. 14% vs. 2%; χ^2^ 51.2, *p* < 0.001). More patients had three or more N2 nodes resected at pneumonectomy, compared to lobar or sublobar resections (98% vs. 85% vs. 38%, respectively; χ^2^ 273.3, *p* < 0.001).

### 3.2. Factors Affecting Nodal Staging Accuracy in Patients with Pre-Operative N1/N2 Staging 

173/973 (18%) patients had pre-operative N1 or N2 suitable for invasive mediastinal staging. While all patients had PET-CT scans performed, among those patients, staging EBUS was performed in 55/173 (32%) patients and mediastinoscopy was performed in 5/173 (3%) patients. In these 173 patients, after adjusting for covariates, younger age (OR 1.05, 95% CI = 1.01–1.09, *p* = 0.02, Table 3) and staging EBUS (OR 2.0, 95% CI = 1.01–4.05, *p* < 0.05) were independent predictors of staging accuracy. In this analysis, sex, performance status, location of the primary tumour, type of surgical procedure, histology, waiting time until surgery and the diagnosis year were not significant on the univariate analysis.

### 3.3. Association of Pre-Operative Nodal Under- and Over-Staging on Survival

Of the patients with concordant pre- and post-operative stages, 81% were alive and 12% died due to lung cancer (Table 1). In the patients with pre-operative under-staging, 54% were alive and 36% died due to lung cancer. In the survival analysis, as expected, the hazard ratio increased with a higher T stage (*p* < 0.001; Table 3). In patients with post-operative N1 and N2 disease, pre-operative nodal under-staging conferred an increased risk of lung cancer-related mortality in comparison to those with accurate pre-operative staging, independent of T stage (N0 concordant as reference; N1 staging concordance HR 1.8, 95%CI 0.8–3.8, *p* = 0.13 vs. N1 nodal under-staging HR 2.9, 95%CI 1.6–5.2, *p* < 0.001; N2 staging concordance HR 2.3, 95%CI 1.02–5.0, *p* = 0.04 vs. N2 nodal under-staging HR 5.2, 95%CI 3.3–8.3, *p* < 0.001; Figure 2). Using adjusted estimates, pre-surgical N2 nodal under-staging had increased mortality in comparison to concordant pre- and post-operative N2 nodal staging (HR 2.33, 95%CI 1.01–5.3, *p* < 0.05). Not receiving adjuvant chemotherapy was associated with a trend suggesting increased mortality (HR 1.6, 95%CI 0.96–2.6, *p* = 0.07). Age, sex, performance status, type of surgical procedure, number of nodes sampled at surgery, time to surgery and pathological subtype had no association with lung cancer-specific mortality on the univariate analysis. There was no association between pre-operative over-staging and lung cancer-specific mortality (Table 3).

## 4. Discussion

Invasive mediastinal staging for radically treatable non-small cell lung cancer (NSCLC) is recommended by all international guidelines, yet its utility remains in question by some respiratory physicians and surgeons. A recent survey of pulmonologists and thoracic surgeons in the USA reported barriers to the application of these guidelines. The predominant reasons for this lack of guideline adherence appearedto be either perceived lack of evidence for systematic staging or inadequate technical expertise [12]. Other barriers to invasive mediastinal staging included potential time delays for additional investigations prior to treatment and institutional reliance on imaging alone for mediastinal staging. Certainly, in this cohort, there was marked variability in practice, with only 35% of patients undergoing surgery with pre-operative N1 or N2 disease and having this confirmed with EBUS or mediastinoscopy, with the majority of patients staged using staging CT and PET-CT alone. These differences in practice may be explained by issues with technical expertise which may in turn influence multidisciplinary team (MDT) decision making with over-reliance on imaging for staging; this would explain the nodal over-staging in this cohort. Whilst MDT meetings allow for shared decision making across specialties in terms of making choices regarding treatment, specialty-specific expertise may influence diagnosis, staging and treatment choices. This heterogeneity may be reduced with regular education, audit of quality performance indicators and external peer review of MDT meetings [13].It is recognized however that there are circumstances in which accurate mediastinal nodal staging is more challenging. In Asian countries, where the proportion of patients with adenocarcinoma is higher both in screened and non-screened populations [14], the sensitivity of PET-CT is lower, with increased specificity in comparison with western countries [15].

We have found that patients with inaccurate pre-operative nodal staging have increased lung cancer-related mortality at 2 years in comparison to those with pre- and post-operative nodal staging concordance. Therefore, our data add to the evidence base supporting invasive staging in patients planned for treatment with radical intent with possible nodal disease on imaging. In addition, with the recent publication and recommendation of neo-adjuvant treatment in patients with stage 1B-3 NSCLC, accurate pre-operative staging is essential to ensure patients with resectable disease receive the optimum treatment. While the introduction of lung cancer screening will result in more early-stage lung cancers being diagnosed, the NELSON study has indicated that there will be a reduction in stage 4 presentations and unchanged proportions of stage 2 and 3 patients who would require mediastinal staging [16].

### 4.1. Accuracy of Pre-Operative Nodal Staging

In this study, the agreement between pre- and post-operative nodal staging was 80%; 13% of patients had a higher post-operative nodal stage, and 7% had unforeseen N2. In a similar study from the Netherlands, Heineman reported very similar findings to those presented in this manuscript using the Dutch Lung Surgery Audit, with an accuracy of nodal staging of 79%, 15% pre-operative nodal under-staging and 6% unforeseen N2, using modern staging techniques [7]. In the era of neo-adjuvant and adjuvant systemic treatment in the context of radical treatment, the importance of pre-operative staging accuracy in confirming (or excluding) nodal involvement prior to surgery will alter a patient’s treatment strategy.

Whilst 89% of patients with pre-operative N0 status had concordant post-operative staging, in the patients with pre-operative N1 and N2, concordant staging was only present in 37% and 58%, respectively. Younger age and the utilisation of staging EBUS were independently associated with staging accuracy. Pneumonectomy was associated with a higher proportion of discordant staging in comparison to lobectomy and sublobar resection. This is important as generally patients with known pre-operative N2 disease are considered unattractive candidates for pneumonectomy. On closer review of the 18 patients upstaged at pneumonectomy, 10 were upstaged from N0 to N1. Of these, all 10 were N1 by direct extension with a N1 node found in the main specimen. Eight were upstaged from N1 to N2, and all eight had occult N2 disease (not FDG-avid). This highlights the importance of systematic staging EBUS, particularly in patients undergoing a planned pneumonectomy. In addition, a higher number of lymph nodes were resected at pneumonectomy in comparison to lobectomy and sublobar resection, thus increasing the likelihood of discovering occult nodal disease. Similar to our cohort, Edwards et al. described that a higher proportion of patients who underwent pneumonectomy had at least three N2 nodal stations sampled, in comparison to lobectomies and sublobar resections [17]. Therefore, pre-operative under-staging may be underestimated in patients undergoing lobectomy and sublobar resection due to less thorough lymph node resection. It is important to note that, although patients undergoing pneumonectomy had proportionally higher-stage disease, neither type of surgery nor number of lymph nodes dissected were associated with increased lung cancer-specific mortality.

Pre-operative over-staging occurs when FDG-avid lymph nodes on PET-CT are not confirmed to be malignant pathologically. The false-positive rate of PET-CT for mediastinal lymph nodes is up to 40%. Consistent with this, in this cohort, there was no increased mortality between pre-operative nodal over-staging and patients with concordant staging, confirming it is not acceptable to rely on the results of the PET-CT alone. Given the gold standard of neo-adjuvant treatment of patients with resectable locally advanced lung cancer, assuming lymph node involvement due to FDG uptake will result in over-staging and patients receiving potentially toxic and expensive pre-operative treatment that is not indicated. In the SEISMIC study, Steinfort et al. reported a reduction in the volume of mediastinal disease in 25% of patients planned for radical radiotherapy who underwent systematic EBUS staging, which resulted in either a smaller volume treated or a switch of treatment to surgery. In addition, occult N2 nodes were discovered in 12% of patients, resulting in a change in treated volume [18]. As such, nodal over-staging will result in patients either being considered for inappropriate neo-adjuvant or adjuvant treatment in the radical treatment setting or could deem patients unsuitable for radical treatment at all. 

### 4.2. Effect of Nodal Staging Accuracy on Lung Cancer-Specific Mortality

We have found that pre-operative under-staging is independently associated with increased risk of lung cancer-specific mortality at two years in comparison with pre-operative concordant nodal staging for the same pathological nodal stage. This is an important finding for several reasons. Principally, it confirms the relevance and importance of the recommendations of international guidelines, specifically regarding the application of invasive mediastinal staging when indicated. 

The increased risk of mortality was present in patients with both unexpected N1 and N2 disease, and on the regression analysis, the effect size was greater in patients with unexpected N2 after adjusting for other relevant variables. Two recent studies from the Netherlands also highlighted the importance of systematic staging to guide appropriate treatment. Bousema et al. found evidence of significant unexpected N2 disease in patients with nodal imaging appearances that would indicate invasive mediastinal staging [19]. In addition, the SCORE study found that systematic mediastinal staging was more effective than targeted staging based on CT and PET-CT appearances [20]. As regards survival, in Lung-BOOST, a post hoc analysis of 133 patients with NSCLC, the group randomized to undergo EBUS-TBNA as their first test had an improvement in overall survival in comparison to patients undergoing conventional diagnosis and staging [21]. However, in a meta-analysis, Navani and colleagues found that there was no independent association between inaccurate clinical TNM staging and all-cause mortality [22]. In the current radical treatment landscape for NSCLC, identifying patients appropriate for neo-adjuvant and adjuvant treatment by determining an accurate stage is essential to improve survival, particularly in patients with locally advanced disease.

The reasons that pre-operative under-staging may be associated with increased mortality remain unclear. One explanation is that patients with disease clearly defined on PET and mediastinal staging are more straightforward to treat in comparison with occult disease that is not delineated using current staging techniques. It is currently not known if patients with occult N2 disease have a poorer prognosis in comparison to patients with clearly evident N2 disease. Alternatively, it is possible that in centres where staging is more thorough, the delivery of treatment may also be more appropriate. Typically, in patients proven to have multi-station N2 disease, patients are more appropriately treated with radical chemoradiotherapy than surgical resection. In addition, if a surgeon is aware of specific nodal involvement prior to resection, then they will be more likely perform a more thorough lymphadenectomy to try and ensure complete clearance of disease.

### 4.3. Strengths and Limitations

One of the major strengths of this study is completeness of data. Consecutive patients diagnosed with lung cancer across multiple hospital sites and treated with surgery in 2015–2017 were included. However, there was a lower than recommended use of invasive staging modalities in our study; this may be representative of the variability of adherence to international guidelines on mediastinal staging outside of large teaching centres. Indeed, this is likely to explain the proportion of patients who were over-staged pre-operatively. This variability did enable us to demonstrate that staging with EBUS-TBNA is an independent predictor of pre- and post-operative intrathoracic nodal staging concordance.

There are well-described limitations of using routine death certificate data for cause-specific mortality. However, using electronic patient records, we were able to establish evidence of lung cancer recurrence in all patients with lung cancer recorded as cause of death, and the non-lung cancer causes of deaths were also reviewed and confirmed. In our study, around a third of patients who died within 2 years of surgery were confirmed to have a cause of death other than lung cancer.

## 5. Conclusions

Mediastinal staging with EBUS was independently associated with pre- and post-operative staging concordance. Pre-operative under-staging was associated with higher risk of lung cancer-specific mortality in comparison to concordant pre-and post-operative nodal staging. Pre-operative nodal staging accuracy in potentially curable non-small cell lung cancer is of fundamental importance to ensure patients receive the correct first-line treatment and to improve survival.

## Figures and Tables

**Figure 1 diagnostics-15-00430-f001:**
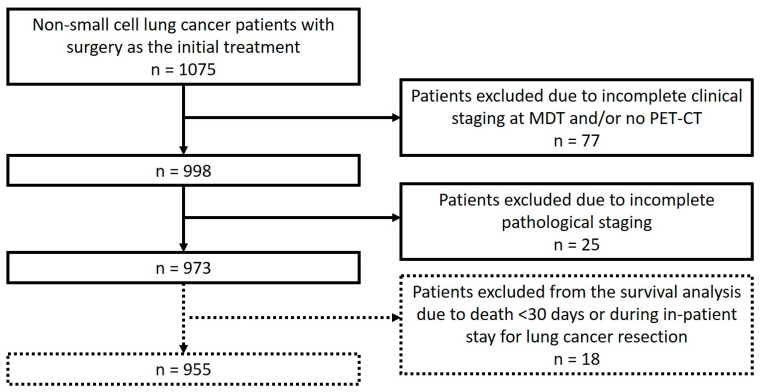
Patients included in the analyses.

**Figure 2 diagnostics-15-00430-f002:**
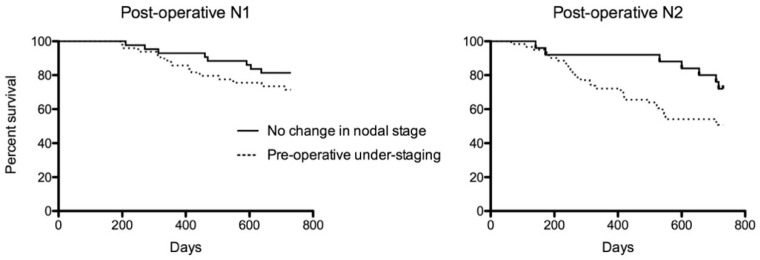
Post-operative survival by nodal staging accuracy.

**Table 1 diagnostics-15-00430-t001:** Patient characteristics.

	Total	Accurate Nodal Staging	Pre-Operative Over-Staging	Pre-Operative Under-Staging
**Number of patients**	973	783 (80%)	67 (7%)	123 (13%)
**Mean age (SD)**	69 (9)	68 (9)	69 (8)	69 (8)
**Sex, n (%)**				
Male	443 (46%)	360 (46%)	27 (40%)	56 (46%)
Female	530 (54%)	423 (54%)	40 (60%)	67 (54%)
**Site of the tumour, n (%)**				
Main bronchus	13 (1%)	7 (1%)	1 (1%)	5 (4%)
Upper lobe	586 (60%)	475 (61%)	39 (58%)	72 (58%)
Middle lobe	39 (4%)	32 (4%)	2 (3%)	5 (4%)
Lower lobe	335 (34%)	269 (34%)	25 (37%)	41 (33%)
**Median (IQR) days from CT to surgery**	74 (56–97)	75(56–98)	69(53–84)	70(54–96)
**Surgical procedure, n (%)**				
Lobectomy	836 (86%)	679 (87%)	58 (87%)	99 (80%)
Pneumonectomy	42 (4%)	18 (3%)	6 (9%)	18 (15%)
Sublobar resection	95 (10%)	86 (11%)	3 (4%)	6 (5%)
**Histology, n (%)**				
Squamous	337 (35%)	261 (33%)	32 (48%)	44 (36%)
Adenocarcinoma	553 (57%)	463 (60%)	29 (43%)	61 (50%)
Other	83 (9%)	59 (8%)	6 (9%)	18 (15%)
**Pre-operative TNM stage, n (%)**				
Stage I	655 (67%)	587 (75%)	0	68 (55%)
Stage II	227 (23%)	146 (19%)	36 (54%)	45 (37%)
Stage III	91 (9%)	50 (6%)	31 (46%)	10 (8%)
**Pre-operative T stage, n (%)**				
T1	528 (54%)	453 (58%)	21 (31%)	54 (44%)
T2	321 (33%)	245 (31%)	28 (42%)	48 (39%)
T3	102 (11%)	72 (9%)	14 (21%)	16 (13%)
T4	22 (2%)	13 (2%)	4 (6%)	5 (4%)
**Pre-operative N stage, n (%)**				
N0	800 (82%)	708 (90%)	0	92 (75%)
N1	119 (12%)	44 (6%)	44 (66%)	31 (25%)
N2	54 (6%)	31 (4%)	23 (34%)	0
**2-year lung cancer mortality, n (%)**				
Alive	752 (77%)	633 (81%)	52 (78%)	67 (54%)
Died due to lung cancer	146 (15%)	91 (12%)	11 (16%)	44 (36%)
Post-operative death	18 (2%)	11 (1%)	2 (3%)	5 (4%)
Died due to other causes	57 (6%)	48 (6%)	2 (3%)	7 (6%)

NSCLC: non-small cell lung cancer.

**Table 2 diagnostics-15-00430-t002:** Agreement between clinical and pathologic nodal stage.

		Post-Operative N Stage				
		pN0	pN1	pN2	Total	Accuracy of Pre-Operative Staging
Pre-operative N stage	N0	708	54	38	800	89%
	N1	44	44	31	119	37%
	N2	15	8	31	54	58%
Total		767	106	100	973	

White: accurately staged; light grey: pre-operative over-staging; dark grey: pre-operative under-staging.

**Table 3 diagnostics-15-00430-t003:** Regression analysis of factors affecting lung cancer-related mortality in patients undergoing surgery for non-small cell lung cancer.

			Hazard Ratio	95% Confidence Interval	*p*-Value
Age			1.01	0.99–1.03	0.28
Sex	Female		1 (ref)		
	Male		1.17	0.83–1.62	0.37
Pathological T stage	T1		1 (ref)		
	T2		1.26	0.83–1.90	0.28
	T3		2.74	1.71–4.38	<0.001
	T4		7.57	4.04–14.17	<0.001
Pathological N stage by post-operative nodal staging concordance	N0	pre-operative stage concordant	1 (ref)		
		pre-operative over-staging	1.17	0.58–2.37	0.65
	N1	pre-operative stage concordant	1.79	0.85–3.78	0.13
		pre-operative over-staging	1.43	0.20–10.36	0.72
		pre-operative under-staging	2.85	1.57–5.18	<0.001
	N2	pre-operative stage concordant	2.26	1.02–4.97	0.04
		pre-operative under-staging	5.24	3.33–8.27	<0.001
Adjuvant chemotherapy	Yes		1 (ref)		
	No		1.58	0.96–2.61	0.07

Performance status, type of surgical procedure, number of nodes sampled at surgery, time to surgery and pathological subtype were not significant.

## Data Availability

The datasets presented in this article are not readily available because they are only available on application to the West of Scotland Cancer Network (WOSCAN). Requests to access the datasets should be directed to the WOSCAN audit team.

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
