# Peer review of "Real-Life Pre-Operative Nodal Staging Accuracy in Non-Small Cell Lung Cancer and Its Relationship with Survival"

_diagnostics, 2025, doi:10.3390/diagnostics15040430_

Round 1
Reviewer 1 Report
Comments and Suggestions for Authors
Dear Author,
the topic is already well known. The invasive preoperative staging might be mandatory. It is not clear which invasive method is more adequated to reach diagnosis. It is not clear why the survival analysis is performed at 2 years.
Author Response
General: We appreciate the reviewer's comments and the predominantly positive feedback on the Review Report Form. We have specifically addressed reviewer 1’s comments.
C1: It is not clear which invasive method is more adequated to reach diagnosis.
R1: While it is unclear whether invasive mediastinal sampling with mediastinoscopy or EBUS is more appropriate for staging in lung cancer, Bousema et al published a prospective study demonstrating that staging EBUS was non-inferior to EBUS plus mediastinoscopy prior to surgery in terms of unforeseen N2 disease (https://doi.org/10.1200/JCO.22.01728). In addition, mediastinoscopy has increased morbidity and mortality in comparison with EBUS. We have added a sentence to paragraph 3 in the introduction regarding this (lines 62-65). This has added a reference which will need to be added to the References section.
C2: It is not clear why the survival analysis is performed at 2 years.
R2: Even in surgically treated NSCLC, recurrence and lung cancer specific mortality are most commonly seen within two years of follow up. We have reported differences between our groups with pre-operative nodal under-staging and pre-operative over-staging or concordant staging at two years. While most studies use overall survival as an outcome measure, we reviewed electronic case records and registered cause of death to use lung cancer specific mortality as a more accurate endpoint. We reported that nodal under-staging was independently associated with lung cancer specific mortality in comparison to concordant or pre-operative over-staging.
Reviewer 2 Report
Comments and Suggestions for Authors
The retrospective study “Real life pre-operative nodal staging accuracy in non-small cell lung cancer and its relationship with survival” is, in general, well-written regarding format and scientific aspects.
My comments:
1. In 55 patients that had staging EBUS-TBNA, were there patients overstaged N3 by PET or CT but staged N0, N1 or N2 by EBUS-TBNA and/or surgery? This should be given in the Abstract and Results, and mentioned and commented on in the Discussion.
2. Although given in Table 3 indirectly, the authors have not mentioned about the lung cancer-specific mortality in patients with preoperative overstaging. This should be given in the Abstract and Results, and should be mentioned and commented on in the Discussion too.
3. Survival curves for the patients with preoperative understaging and overstaging should be given.
4. The Results and Discussion should be separated into subtopics so that the reader can more easily read and understand many data and related comments and comparisons with the pertinent literature.
Author Response
General: We appreciate the reviewer's comments and the positive feedback on the Review Report Form. I have specifically addressed reviewer 2’s comments.
C1. In 55 patients that had staging EBUS-TBNA, were there patients overstaged N3 by PET or CT but staged N0, N1 or N2 by EBUS-TBNA and/or surgery? This should be given in the Abstract and Results, and mentioned and commented on in the Discussion.
R1: Patients who were staged pre-operatively as N3 with EBUS or PET-CT would not have been referred for surgery and as such would not have been included in this study. In the patients and methods section, we have commented that ‘In general, only patients with N0, N1 or single station N2 would be considered for surgery’ (line 100-101). Of the 55 patients who had EBUS-TBNA, their pre-operative stage will have been determined by both PET-CT and staging EBUS combined - some may have had N3 lymph nodes with FDG uptake exonerated with EBUS, and as such their pre-operative stage would have been N1 or N2 and rendered them suitable for surgery.
C2. Although given in Table 3 indirectly, the authors have not mentioned about the lung cancer-specific mortality in patients with preoperative overstaging. This should be given in the Abstract and Results, and should be mentioned and commented on in the Discussion too.
R2: As recognised by the reviewer, the lung cancer-specific mortality for the pre-operative nodal over-staged cohort is not different to the patients with concordant pre- and post-operative staging. This is evident from Table 1 (2-year lung cancer mortality) and Table 3 (HR for overstaging is not different to concordant staging).
We have mentioned in the results section that 'There was no association between pre-operative over-staging and lung cancer-specific mortality’ (lines 166-167). I have added a reference to Table 3 after this sentence. We have added ‘There was no increased mortality associated with pre-operative nodal over-staging' to the abstract (lines 29-30). We have also added ‘Consistent with this, in this cohort, there was no increased mortality between pre-operative nodal over-staging and patients with concordant staging, confirming it is not acceptable to rely on the results of the PET-CT alone.’ to the discussion (lines 234-236).
C3. Survival curves for the patients with preoperative understaging and overstaging should be given.
R3: We have included a survival curve (Figure 2) and referred to it in the results section (line 165).
Figure 2. Post-operative survival by nodal staging accuracy
See attachment
C4. The Results and Discussion should be separated into subtopics so that the reader can more easily read and understand many data and related comments and comparisons with the pertinent literature.
R4: We have added titles to both the results and the discussion section as suggested to make it more straightforward for the reader.

Round 2
Reviewer 1 Report
Comments and Suggestions for Authors
Dear authors,
thank you for your reply.
Author Response
There are no further comments to reply to.
Reviewer 2 Report
Comments and Suggestions for Authors
Dear Authors,
Your replies to my comments and the revisÅŸon of your paper is satisfactory.
Author Response

(The authors gave the same response as above.)
